# Exploration of the metabolomic mechanisms of postmenopausal hypertension induced by low estrogen state

Yao Li[1], Hui Xin[1], Zhexun Lian[1]*, Wei Zhang[2]*

[1]Department of Cardiology, The Affiliated Hospital of Qingdao University, Qingdao, China; [2]Department of Urology, The Affiliated Hospital of Qingdao University, Qingdao, China

## eLife Assessment

This **useful** study provides **incomplete** evidence regarding the pathophysiological role of low estrogen levels post-menopause in hypertension, focusing on L-AABA as a key mediator. The results describe a novel hypothesis for the pathophysiology of hypertension in this population and are of interest to experts in hypertension and vascular biology.

*For correspondence:
Lianzx566@163.com (ZL);
1711110480@pku.edu.cn (WZ)

**Abstract** Estrogen significantly impacts women's health, and postmenopausal hypertension is a common issue characterized by blood pressure fluctuations. Current control strategies for this condition are limited in efficacy, necessitating further research into the underlying mechanisms. Although metabolomics has been applied to study various diseases, its use in understanding postmenopausal hypertension is scarce. Therefore, an ovariectomized rat model was used to simulate postmenopausal conditions. Estrogen levels, blood pressure, and aortic tissue metabolomics were analyzed. Animal models were divided into Sham, OVX, and OVX +E groups. Serum estrogen levels, blood pressure measurements, and aortic tissue metabolomics analyses were performed using radioimmunoassay, UHPLC-Q-TOF, and bioinformatics techniques. Based on the above research content, we successfully established a correlation between low estrogen levels and postmenopausal hypertension in rats. Notable differences in blood pressure parameters and aortic tissue metabolites were observed across the experimental groups. Specifically, metabolites that were differentially expressed, particularly L-alpha-aminobutyric acid (L-AABA), showed potential as a biomarker for postmenopausal hypertension, potentially exerting a protective function through macrophage activation and vascular remodeling. Enrichment analysis revealed alterations in sugar metabolism pathways, such as the Warburg effect and glycolysis, indicating their involvement in postmenopausal hypertension. Overall, this current research provides insights into the metabolic changes associated with postmenopausal hypertension, highlighting the role of L-AABA and sugar metabolism reprogramming in aortic tissue. The findings suggest a potential link between low estrogen levels, macrophage function, and vascular remodeling in the pathogenesis of postmenopausal hypertension. Further investigations are needed to validate these findings and explore their clinical implications for postmenopausal women.

## Introduction

Estrogen has a multifaceted impact on women, with postmenopausal hypertension being a representative example, characterized by significant fluctuations in blood pressure, primarily in systolic and pulse pressure. This leads to more damage to target organs, ultimately severely affecting the quality

**Figure 1.** Research flowchart.

of life of middle-aged and elderly women (**Benjamin et al., 2017**; **Rossi et al., 2011**). Doctors have attempted to improve patients' conditions through hormone replacement therapy, but the serious risk of breast cancer has greatly limited the clinical application of this treatment (**Fournier et al., 2009**). Angiotensin-converting enzyme inhibitors (ACEI) or angiotensin receptor blockers (ARB) in combination with beta-blockers or verapamil are common control strategies for postmenopausal hypertension, yet satisfactory therapeutic effects are still difficult to achieve (**Chen et al., 2023**). Research has long confirmed the influence of estrogen on vascular pressure, for instance, oxidative stress caused by estrogen metabolism changes may result in pulmonary arterial hypertension in obese patients (**Xu et al., 2021**). The detailed mechanism by which the drastic decrease in estrogen post-menopause mediates postmenopausal hypertension remains unclear. In recent years, metabolomics strategies have been widely used to explore potential mechanisms of various diseases, with plasma metabolomics being the most popular (**Ferraro et al., 2024**). Some researchers have utilized plasma metabolomics to investigate the pathogenesis of thrombotic or idiopathic pulmonary arterial hypertension (**Pi et al., 2023**; **Mey et al., 2020**; **Swietlik et al., 2021**), while others have described the plasma metabolomic characteristics of high blood pressure induced by high sodium intake (**Rossitto et al., 2021**). There has been no research exploring the metabolic changes in postmenopausal hypertension yet. Our research team conducted this study to address the aforementioned scientific issues. By simulating the postmenopausal state using an ovariectomized rat model, we monitored the correlation between estrogen and blood pressure changes in rats and revealed the inherent relationship between these changes and postmenopausal hypertension through alterations in the metabolic characteristics of aortic tissues *Figure 1*.

## Results

### Development of animal models with estrogen depletion

In the Sham group, normal estrous cycle patterns were observed, with small, round vaginal exfoliated cells present during the nonestrous phase and larger, polygonal cells with abundant cytoplasm seen during the estrous phase. Conversely, in the OVX and OVX + E groups (pre-estrogen supplementation), the estrous cycle ceased, and the vaginal exfoliated cells remained small and round, confirming the effectiveness of bilateral ovariectomy. Serum estrogen levels in the OVX group were significantly

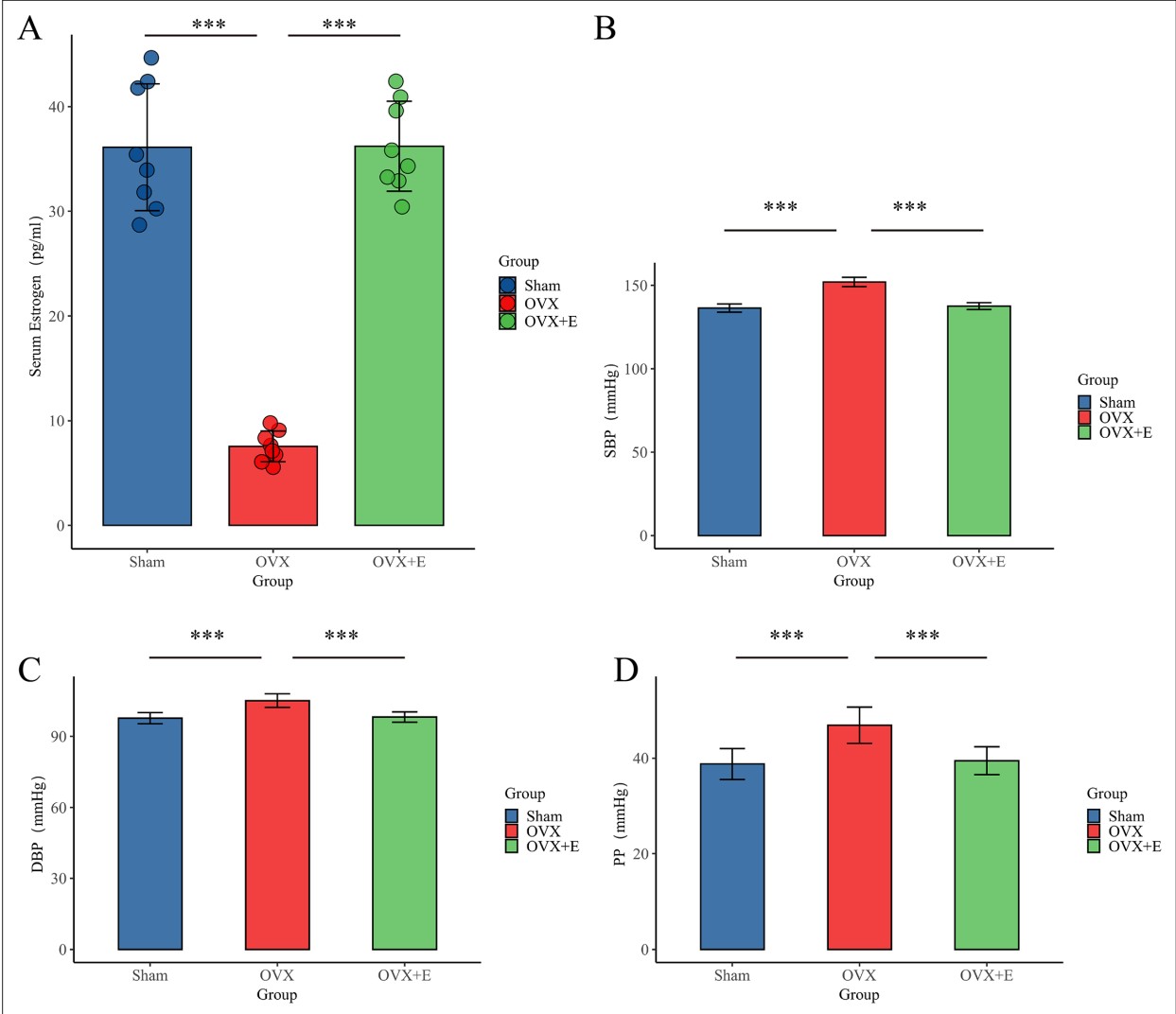

**Figure 2.** Establishment of animal Models. (**A**) Serum estrogen concentration, n=8 per group (**B**). SBP of animal, n=8 per group (**C**) DBP, n=8 per group. (**D**) PP of animal, n=8 per group. ***p<0.001.

lower compared to the Sham group (7.54 ± 1.46pg/mL vs. 36.12 ± 6.07pg/mL, n = 8, p< 0.001), while levels in the OVX + E group were markedly higher than those in the OVX group (36.21 ±4.30pg/mL vs. 7.54 ± 1.46pg/mL, n = 10, p< 0.001). There was no significant difference between the Sham and OVX + E groups, further confirming the validity of the experimental model (*Figure 2A*).

## BP features of the OVX model

The SBP (151.98±2.79 mmHg), DBP (105.10±2.89 mmHg), and PP (46.88±3.78 mmHg) of the OVX group were significantly higher than the other two groups (p<0.001). There was no statistical difference between the Sham group and the OVX + E group in SBP (136.43±2.45 mmHg vs. 137.60±2.03 mmHg), DBP (97.65±2.34 mmHg vs. 98.13±2.16 mmHg), and PP (38.78±3.24 mmHg vs. 39.48±2.92 mmHg) (p>0.05; *Figure 2B–D*).

## Characteristics of aortic metabolites related to low estrogen levels

A total of 184 metabolites in aortic tissues were identified using metabolomics analysis (*Supplementary file 1*: Metabolomics raw data), mainly categorized into 8 major classes, with amino acids accounting for over 1/3 (*Figure 3A*). To normalize the data distribution, both metabolites and samples underwent normalization procedures (*Figure 3—figure supplement 1A, B*). To explore the differences in aortic tissue metabolites under varying estrogen levels, a one-way analysis of variance

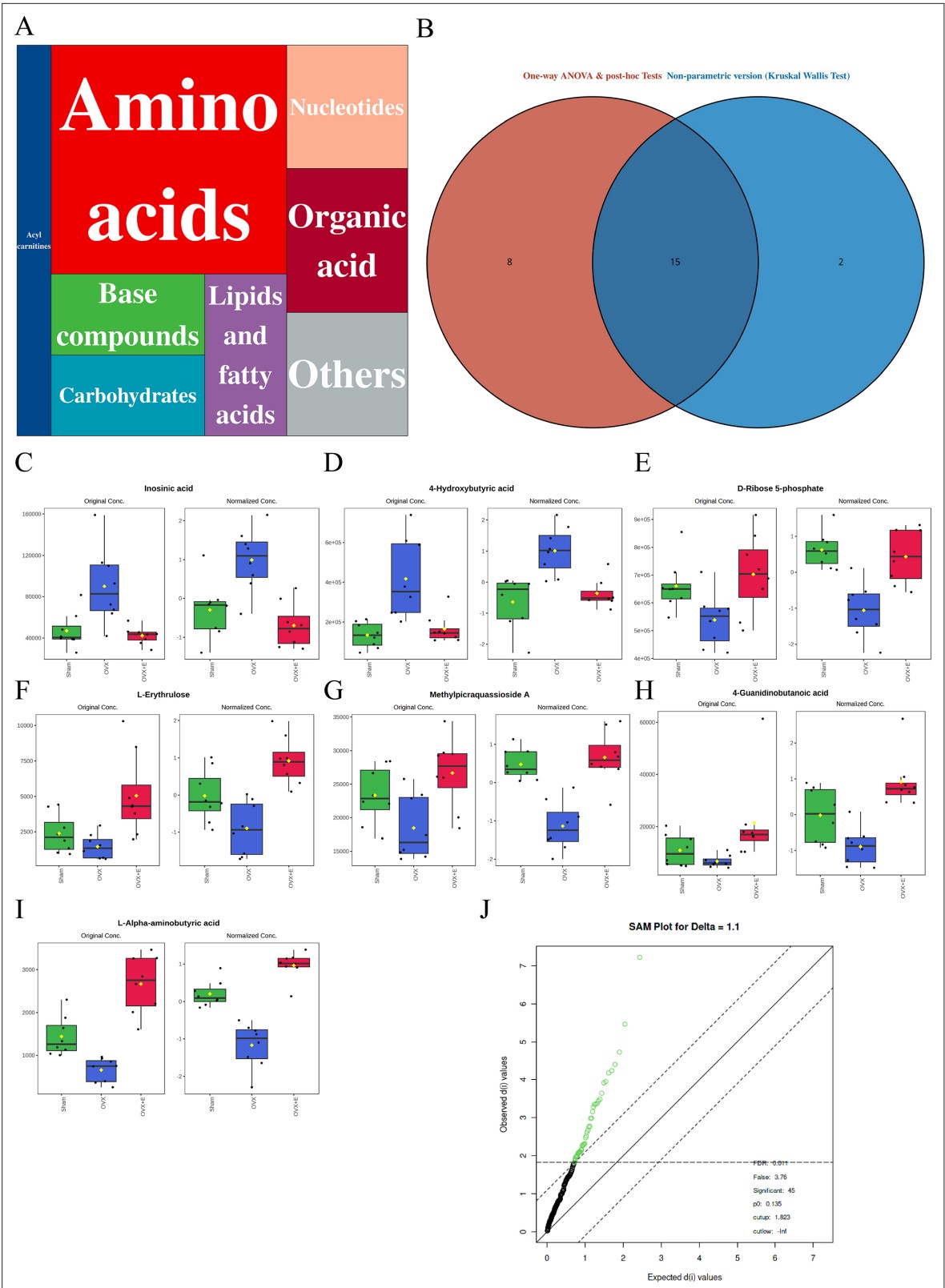

**Figure 3.** Overview of aorta metabolomics related to estrogen deprivation. (**A**) Metabolite classification statistics chart. (**B**) Venn diagram of differential metabolites. (**C–I**) Bar chart of the expression levels of the most promising differential metabolites. (**J**) SAM analysis of differential metabolites.

The online version of this article includes the following figure supplement(s) for figure 3:

*Figure 3 continued on next page*

*Figure 3 continued*

**Figure supplement 1.** Normalization processing rendering.

**Figure supplement 2.** Multi-class significance analysis model.

**Figure supplement 3.** A heatmap of metabolite correlations.

(ANOVA) was utilized to select differentially expressed metabolites among the three groups of aortas. The false discovery rate threshold was set at 0.05. Since the normalized data approximated a normal distribution, hypothesis testing was carried out through two methods: non-parametric tests identified 17 different metabolites (*Supplementary file 2*: Non-parametric tests of metabolites), while Fisher's Least Significant Difference (LSD) tests revealed 23 different metabolites (*Supplementary file 3* : Fisher's LSD tests of metabolites). By taking the intersection of the two methods, a total of 15 different metabolites were identified (*Supplementary file 4* : Intersection differential metabolites). Further selection of the most promising differentially expressed metabolites among groups was shown in *Figure 3C–I*. Using multi-class significance analysis of microarrays (SAM) (with a delta set at 1.1, *Figure 3—figure supplement 2*), a total of 45 different metabolites were identified (*Figure 3J*, *Supplementary file 5* : Differential metabolites identified by SAM), with L-Alpha-aminobutyric acid (L-AABA), Methylpicraquassioside A, Pyroglutamine, and D-Ribose 5-phosphate being the most promising inter-group differentially expressed metabolites. A heatmap of metabolite correlations indicated that all metabolites could be roughly categorized into 4 distinct clusters based on their correlation relationships (*Figure 3—figure supplement 3*, *Supplementary file 6*, *Supplementary file 7* : The correlation coefficient and p-value of the correlation analysis for all metabolites). The Pearson correlation analysis for the three different sample groups clearly demonstrated distinct differences between the OVX group and the other two groups, while the Sham and OVX + E groups with similar estrogen concentrations were challenging to differentiate (*Figure 4A*, *Supplementary file 8*, *Supplementary file 9* : The correlation coefficient and p-value of the correlation analysis for all samples). The trends highlighted in the hierarchical clustering dendrogram further emphasized these distinctions (*Figure 4B*). The hierarchical clustering heatmap, created based on the top 25 differentially expressed metabolites filtered by ANOVA, depicted two distinct patterns of metabolite expression across different groups. A class of metabolites represented by Inosinic acid showed a significant upregulation in the OVX group, while displaying consistent downregulation in the Sham and OVX + E groups; another class of metabolites primarily displayed downregulation in the OVX group (*Figure 4C*). Further exploration of the expression characteristics of aortic metabolites in the OVX group was performed using dimensionality reduction analyses. Principal Component Analysis (PCA), Partial Least Squares Discriminant Analysis (PLS-DA), and sparse PLS-DA (sPLS-DA) were applied as the three analytical strategies. In the Unsupervised strategy, the first component accounted for 21.5% and the second component for 15%, with a high degree of overlap among the three groups' data (*Figure 5A*, *Figure 5—figure supplement 1A-B*). Implementing the PLS-DA strategy distinguished the differences between the OVX group and the other two groups clearly, although there was still some minor overlap (*Figure 5B*, *Figure 5—figure supplement 1C*). Although the model could select some differentially expressed metabolites using VIP scores (*Figure 5—figure supplement 1D*), a fivefold cross-validation (CV) indicated suboptimal values for $R^2$, $Q^2$, and accuracy indicators (*Figure 5—figure supplement 1E*, *Supplementary file 10* : PLS-DA CV details). The permutation test also suggested a risk of model overfitting (*Figure 5—figure supplement 1F*). As the predictive performance of both the PCA and PLS-DA models fell short of expectations, the sPLS-DA model was tested, showing optimal performance in the fivefold CV results (*Figure 5C*, *Figure 5—figure supplement 1G–H*). According to the VIP scores, L-AABA and Methylpicraquassioside A were identified as the top differentially expressed metabolites (*Figure 5D*, *Supplementary file 11* : VIP scores for differential metabolites). Similarly, in the random forest tree model, L-AABA was identified as the most important metabolite for classification accuracy evaluation (Mean Decrease accuracy; *Figure 5E*, *Supplementary file 12* : Mean Metabolic Accuracy of differential metabolites in random forest tree model). Unfortunately, the out-of-bag (OOB) error for the random forest tree model did not reach 0 (0.0417, *Figure 5F*).

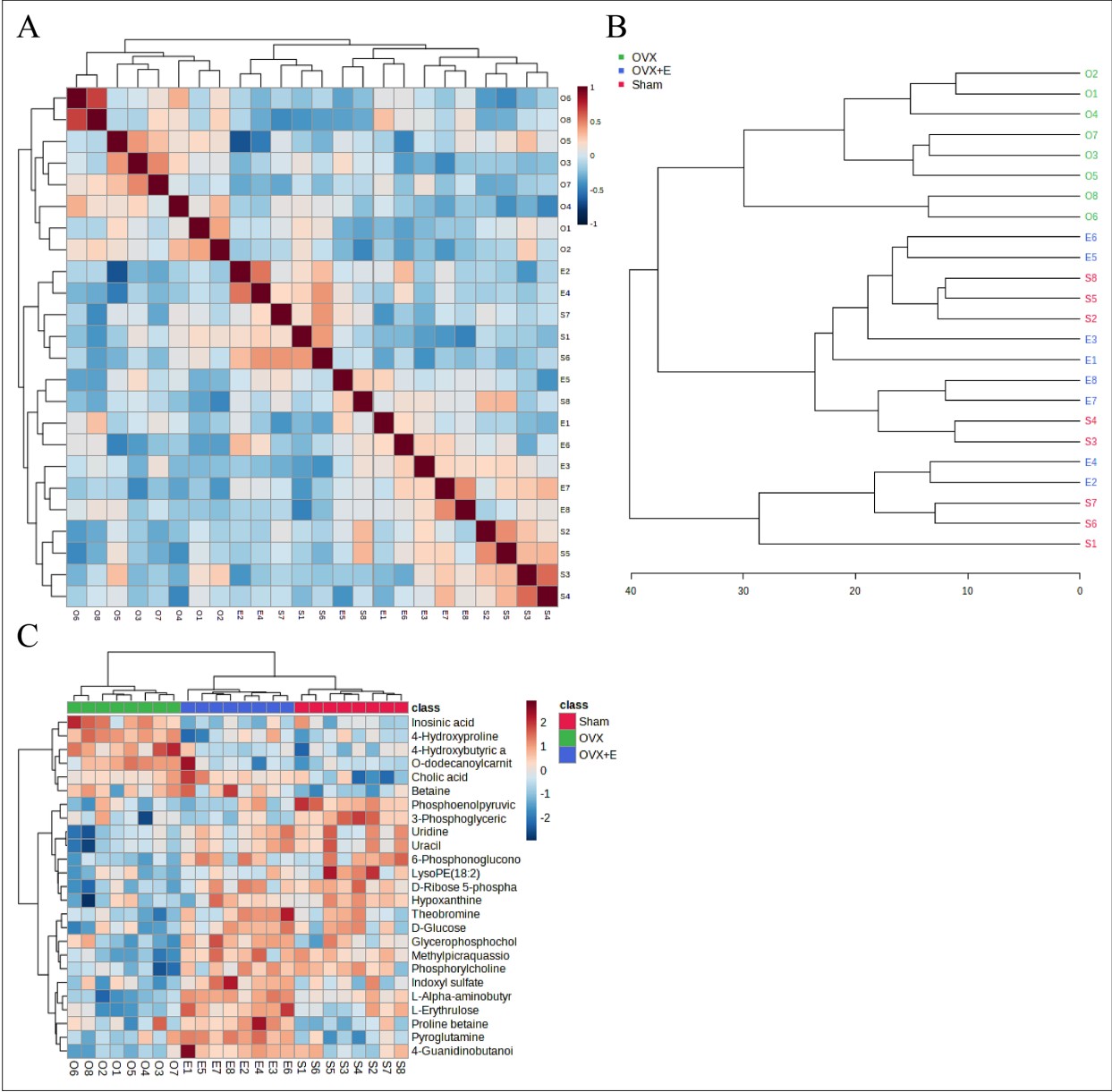

**Figure 4.** Correlation analysis of samples and differential metabolites. (**A**) Pearson correlation analysis heatmap of samples. (**B**) Hierarchical clustering tree diagram of samples. (**C**) Hierarchical clustering heatmap of samples and metabolites.

## Subgroup analysis of metabolic characteristics

Using the aforementioned methods, we obtained overall differences in three sets of aortic metabolites, and further verification is needed to determine whether these differences exhibit the expected trends among different groups. To this end, we comprehensively employed various methods to assess the differential features of metabolites between OVX and the other two groups (t-test, PCA, PLS-DA and Orthogonal PLS-DA (OPLS-DA), Random Forest, and Empirical Bayesian Analysis of Metabolomics (EBAM)). When comparing data of the subgroups, we recalibrated the data (*Figure 6—figure supplement 1A–D*). The Fold Change (FC) threshold for the t-test was set to 2, and the p-value was 0.05. Compared to the Sham group, in the OVX group, Adenylsuccinic acid, 4-Hydroxybutyric acid, Cholic acid, O-dodecanoylcarnitine, L-Hexanoylcarnitine, Adenosine 3′-monophosphate, 1-Methylguanine, O-decanoyl-L-carnitine, and Butyrylcarnitine were significantly upregulated, while Uracil, Uridine, Ribothymidine, and L-AABA were significantly downregulated (*Figure 6A*, *Supplementary file 13* : Details of t-test for differential metabolites between Sham group and OVX group).

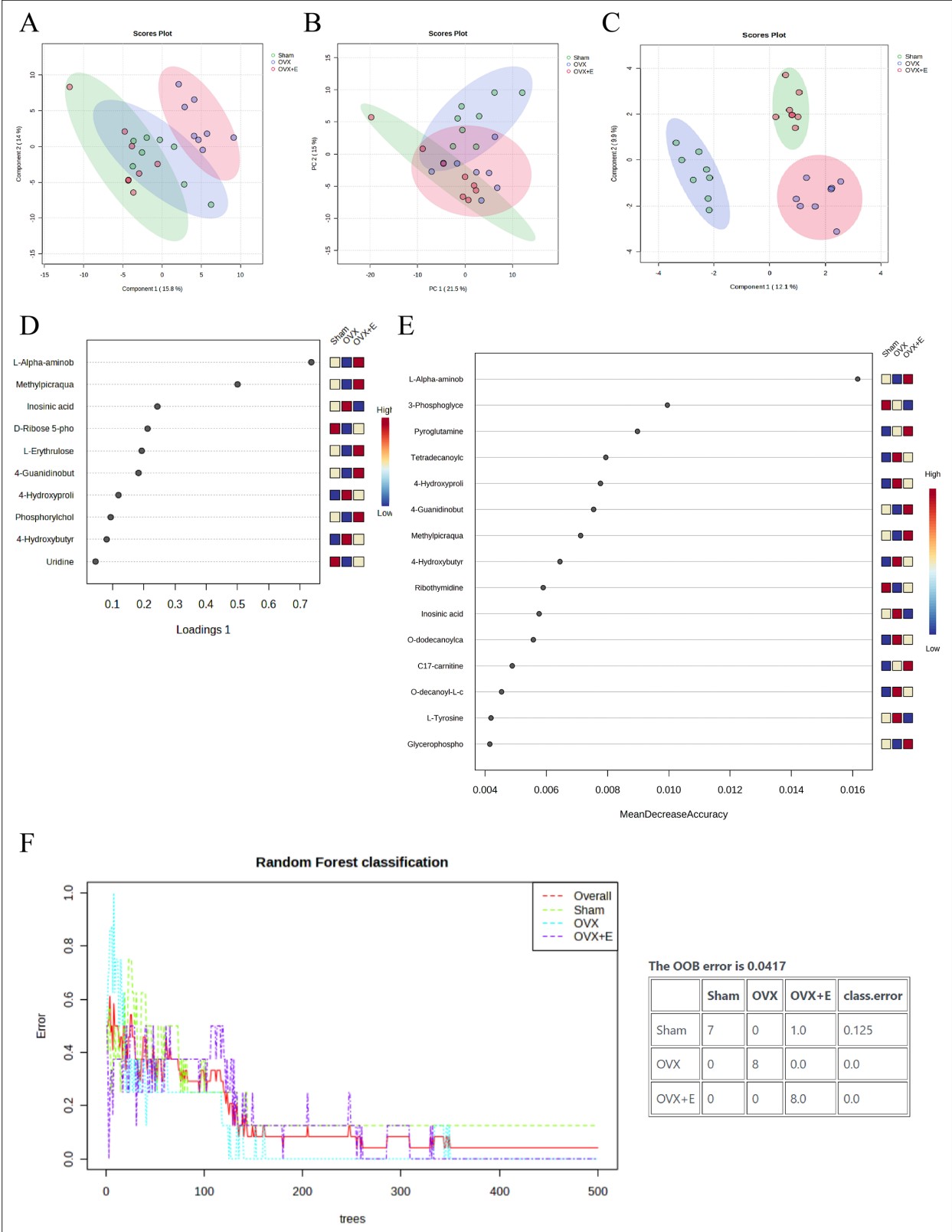

**Figure 5.** Overview of dimension reduction analysis. (**A**) PCA scatter plot. (**B**) PLS-DA scatter plot. (**C**) sPLS-DA scatter plot. (**D**) VIP score of sPLS-DA model. (**E**) VIP score of RF model (**F**) OOB error for the random forest tree model.

The online version of this article includes the following figure supplement(s) for figure 5:

**Figure supplement 1.** Dimension reduction analysis of the three groups.

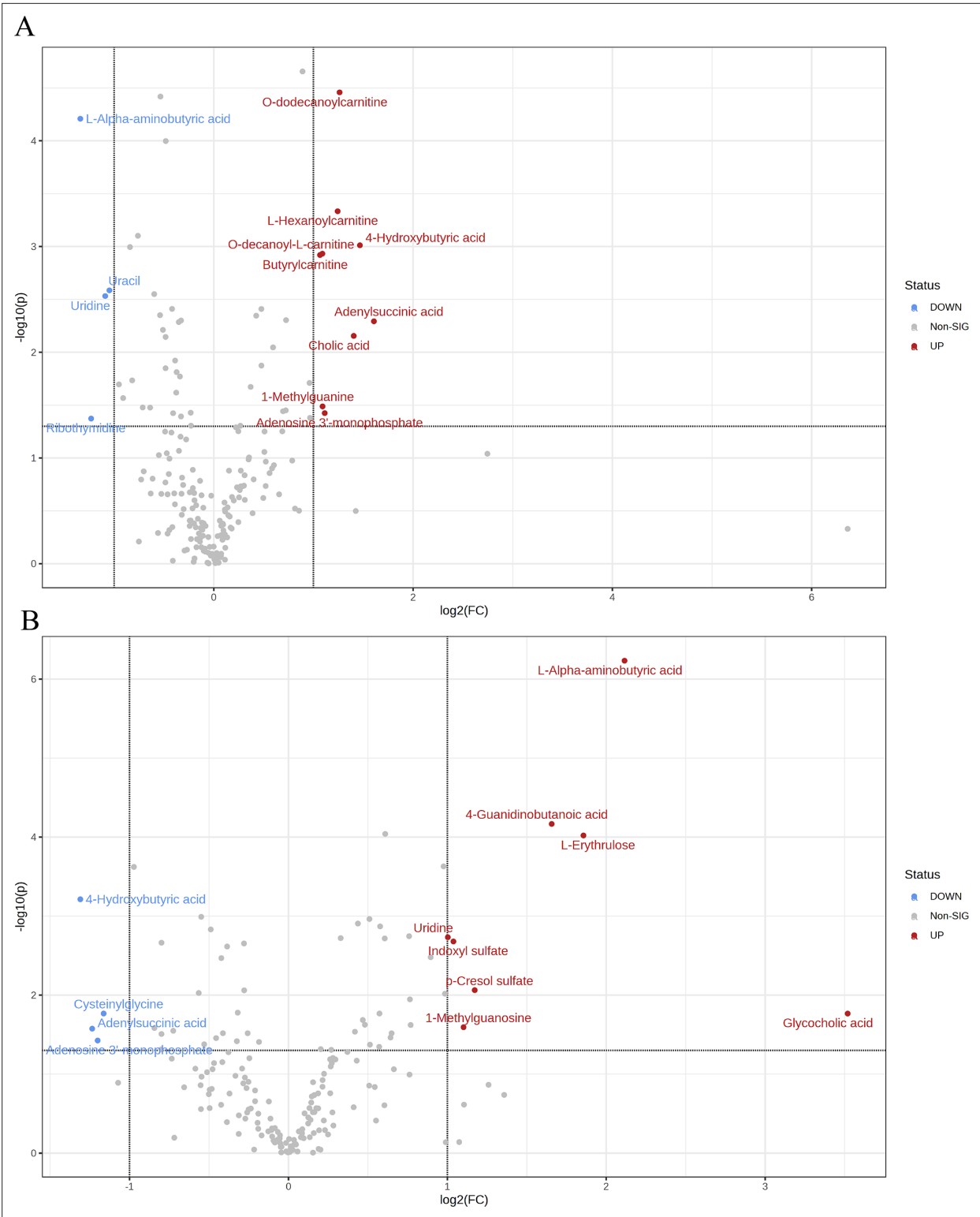

**Figure 6.** Overview of subgroup dimensionality reduction analysis 1. (**A**) Volcano map of differential metabolites between the Sham group and OVX group. (**B**) Volcano map of differential metabolites between the OVX group and OV +E group.

The online version of this article includes the following figure supplement(s) for figure 6:

**Figure supplement 1.** Subgroup normalization processing rendering.

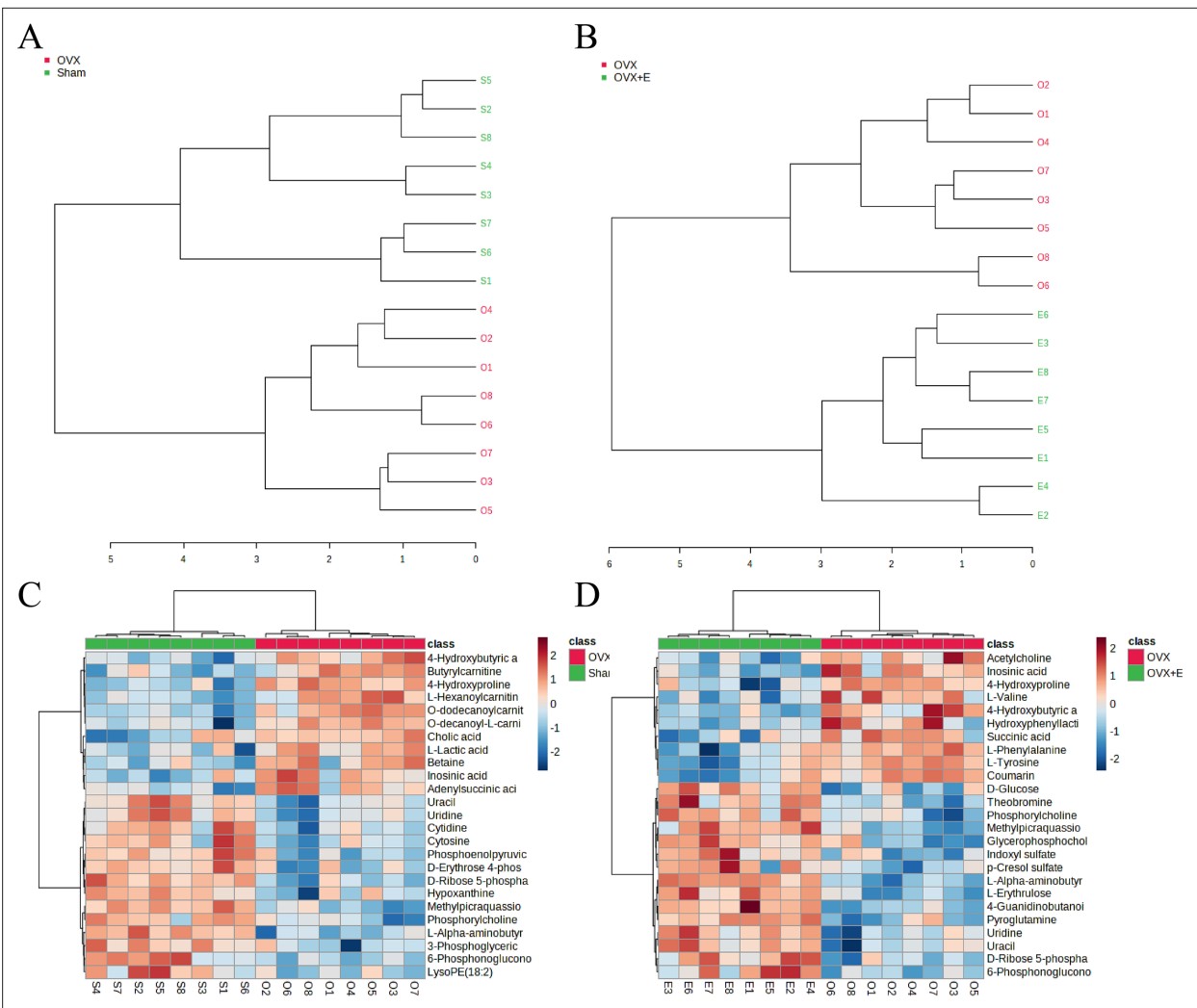

**Figure 7.** Overview of subgroup dimensionality reduction analysis 2. (**A**) Hierarchical clustering dendrogram between the Sham group and OVX group. (**B**) Hierarchical clustering dendrogram between the OVX group and OVX + E group. (**C**) Hierarchical clustering heatmap between the Sham group and OVX group. (**B**) Hierarchical clustering heatmap between the OVX group and OVX + E.

Compared to the OVX group, Cysteinylglycine, Adenosine 3'-monophosphate, Adenylsuccinic acid, and 4-Hydroxybutyric acid were significantly downregulated in the OVX + E group, while Glycocholic acid, L-AABA, L-Erythrulose, 4-Guanidinobutanoic acid, p-Cresol sulfate, 1-Methylguanosine, Indoxyl sulfate, and Uridine were significantly upregulated (*Figure 6B*, *Supplementary file 14* : Details of t-test for differential metabolites between OVX + E group and OVX group). The hierarchical clustering dendrogram clearly distinguished the samples of different subgroups (*Figure 7A–B*). The hierarchical clustering heatmap visually displayed the above results (*Figure 7C–D*). Although PCA analysis as an unsupervised analysis could effectively differentiate between the OVX and OVX + E groups, it faced difficulties in distinguishing differences between the OVX and Sham groups (*Figure 8A-B*, *Figure 8— figure supplement 1A-B*). In contrast, supervised methods were more adept at discerning intergroup differences, but these models also had certain limitations. The PLS-DA model had good intergroup discriminative ability (*Figure 8C-D*, *Figure 8—figure supplement 1C-D*), but showed clear signs of model overfitting (Figure S6E-F). The OPLS-DA model also had advantages in discriminative ability (*Figure 8E–F*), and the permutations test results indicated that the $R^2Y$ values of the OPLS-DA models between OVX and the other two groups were 0.981 (p=0.009) and 0.986 (p=0.001), with $Q^2$ values of 0.848 (p<0.002) and 0.893 (p=0.001; *Figure 8G–H*). The cross-validation results were consistent with the permutations test (*Figure 8—figure supplement 1G–H*), demonstrating the high predictive value of the models. Utilizing the VIP scores and S-Plot of this model, we further identified four important

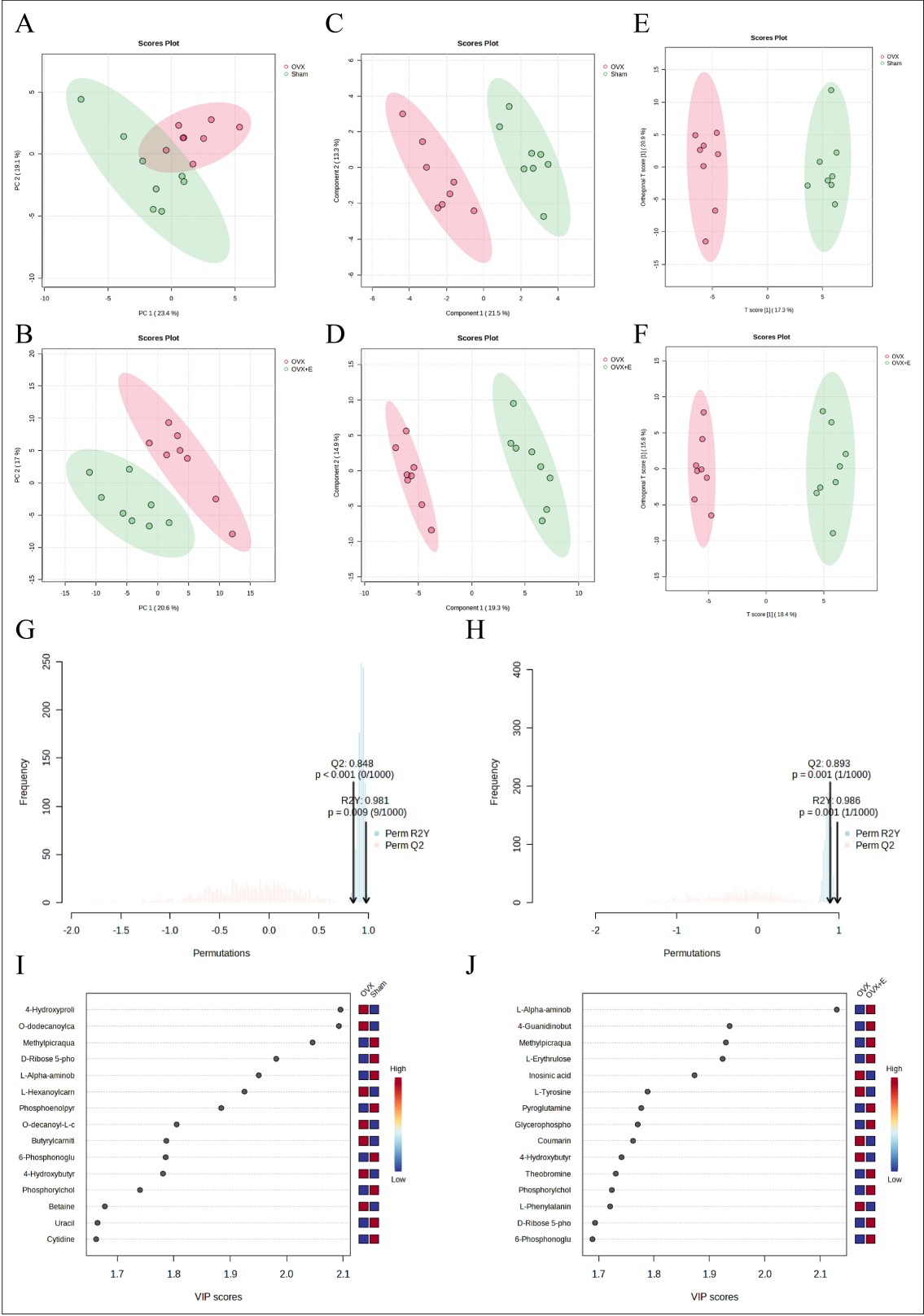

**Figure 8.** The subgroup dimensionality reduction analysis. (**A**) PCA scatter plot between the Sham group and OVX group. (**B**) PCA scatter plot between the OVX group and OVX + E group. (**C**) PLS-DA scatter plot between the Sham group and OVX group. (**D**) PLS-DA scatter plot between the OVX group and OVX + E group. (**E**) OPLS-DA scatter plot between the Sham group and OVX group. (**F**) OPLS-DA scatter plot between the OVX group and OVX + E group. (**G**) OPLS-DA model permutations test between the Sham group and OVX group. (**H**) OPLS-DA models permutations test between the OVX

*Figure 8 continued on next page*

*Figure 8 continued*

group and OVX + E group. (**I**) VIP score of OPLS-DA model between the Sham group and OVX group. (**J**) VIP score of OPLS-DA model between the OVX group and OVX + E group.

The online version of this article includes the following figure supplement(s) for figure 8:

**Figure supplement 1.** The subgroup dimensionality reduction analysis.

**Figure supplement 2.** The subgroup random forest model.

differential metabolites, namely L-AABA, 4-Hydroxyproline, O-dodecanoylcarnitine, and Methylpicraquassioside A (*Figure 8I–J*, *Supplementary file 15*, *Supplementary file 16* : VIP scores of OPLS-DA models between subgroups). The Random Forest model exhibited strong advantages in determining intergroup differences, with an OOB error of 0 (*Figure 8—figure supplement 2A–B*), and the VIP plots based on contribution to classification accuracy provided several promising differential metabolites (*Figure 9A–B*, *Supplementary file 17*, *Supplementary file 18* : VIP scores of the random forest model between subgroups), which still included L-AABA. The evaluation results of EBAM and SAM are shown in *Figure 9C–F* (*Supplementary file 22* : The evaluation details of EBAM and SAM model between subgroups).

## Metabolite expression characteristics

Metabolite expression characteristics are another analytical strategy to explore the impact of low estrogen on the aorta. In the subgroup analysis, using the OVX group as a control, representative metabolites associated with the Sham group and OVX + E group are shown in *Figure 10A–B*; *Supplementary file 23*, *Supplementary file 24*: Details of intergroup differential analysis of low estrogen related metabolites. Among them, the metabolites with a positive correlation coefficient exceeding 0.8 include Methylpicraquassioside A, L-AABA, and D-Ribose 5-phosphate (Sham group), as well as L-AABA, 4-Guanidinobutanoic acid, Methylpicraquassioside A, and L-Erythrulose (OVX +E group). The trend arranged from low to high estrogen concentrations (OVX-Sham-OVX+E) is shown in *Figure 10*, *Supplementary file 25* : Details of metabolite related trends from low to high estrogen concentrations. L-AABA exhibits a significant positive correlation feature (correlation coefficient of 0.89), while Inosinic acid is the only metabolite with a negative correlation coefficient exceeding –0.7 (correlation coefficient of –0.7). Therefore, we further explored metabolites with correlated expression to L-AABA (*Figure 10D*, *Supplementary file 26*: Correlation coefficient of metabolites related to L-AABA expression), where L-Erythrulose shows the most positive correlation (correlation coefficient of 0.87171), and Inosinic acid shows the most negative correlation (correlation coefficient of –0.62298).

## Identification of biomarkers

Univariate receiver operating characteristic (ROC) curve analysis was used to screen promising biomarkers, and L-AABA was found to be the most promising biomarker, with an AUC of 1 in the comparison process between OVX and the other two groups (*Figure 11A–B*, *Supplementary file 27*, *Supplementary file 28*: The promising differential metabolites AUC results of ROC curves between subgroups). Multivariate ROC curve analysis demonstrated unique value in biomarker selection, with the AUC consistently above 0.93 starting from 5 variables (*Figure 11C–D*). The error classification of the multivariate ROC curve showed that there were no misclassifications when using the model for samples in the OVX and OVX +E groups, but one sample from the Sham group was incorrectly classified into the OVX group (*Figure 11—figure supplement 1A–B*). In this multivariate model, L-AABA still ranked in the top three in terms of average importance (*Figure 11E–F*).

## Enrichment analysis of differential metabolites

Further enrichment analysis of differential metabolites was conducted to elucidate the underlying mechanisms. Differential metabolites showed a wide range of classifications, with amino acids and peptides, fatty acids and conjugates, and monosaccharides ranking among the top three (*Figure 12A*). Results of metabolic pathway enrichment analysis indicated that the Warburg effect, glycolysis, and gluconeogenesis were the top three enriched metabolic pathways (*Figure 12B*). Enzyme-specific metabolic analysis revealed that deoxyuridine phosphorylase (UPP) and $O^2$ transport (diffusion) ranked in the top two (*Figure 12C*). To eliminate the impact of species differences on enrichment results, we

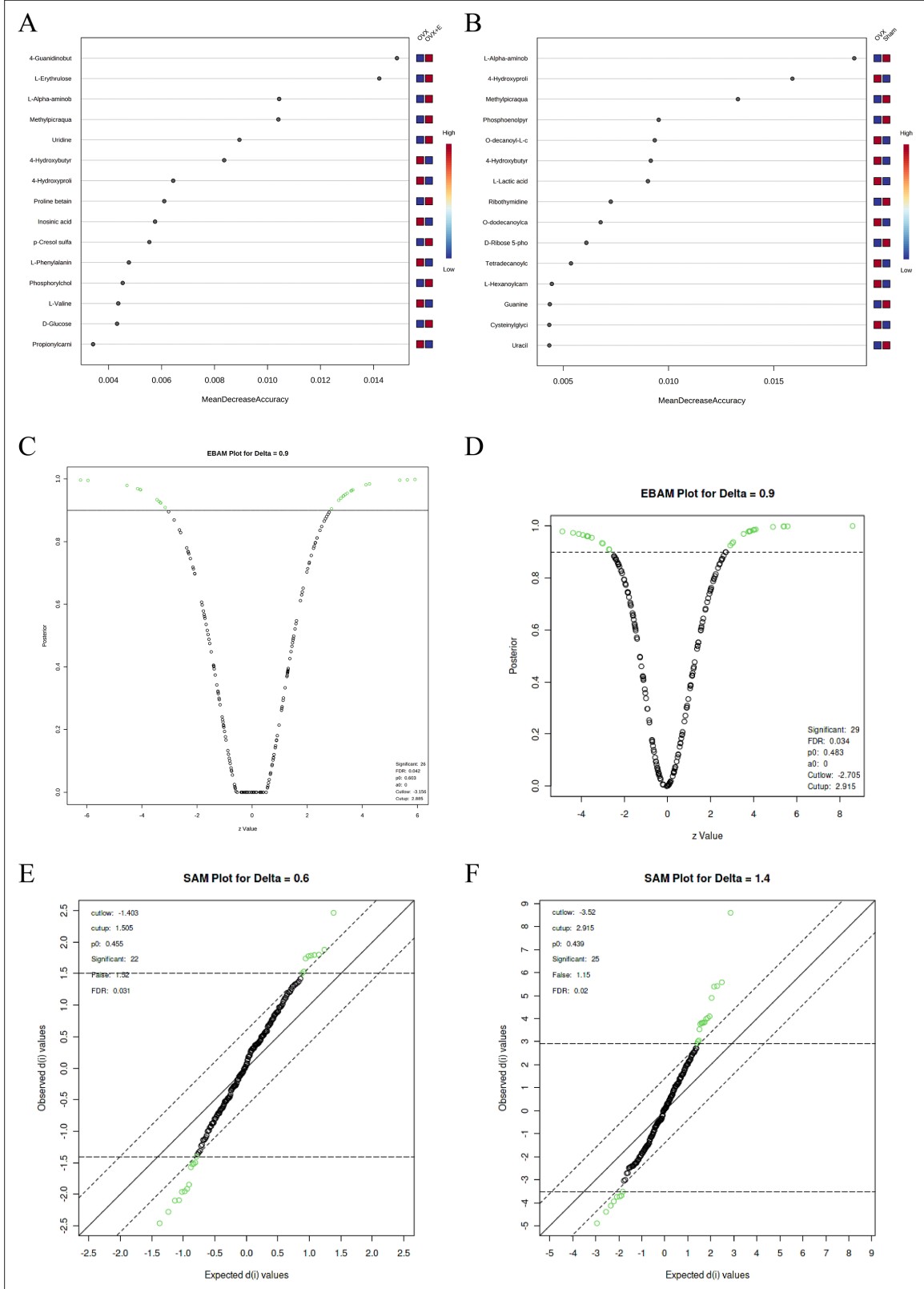

**Figure 9.** Screening differential metabolites using RF, EBAM, and SAM methods. (**A**) VIP patterns based on RF model between the Sham group and OVX group. (**B**) VIP patterns based on RF model between the OVX group and OVX +E group. (**C, D**) Volcano plots for subgroup comparison based on EBAM method. (**E, F**) Screening of differential metabolites between subgroups based on SAM method.

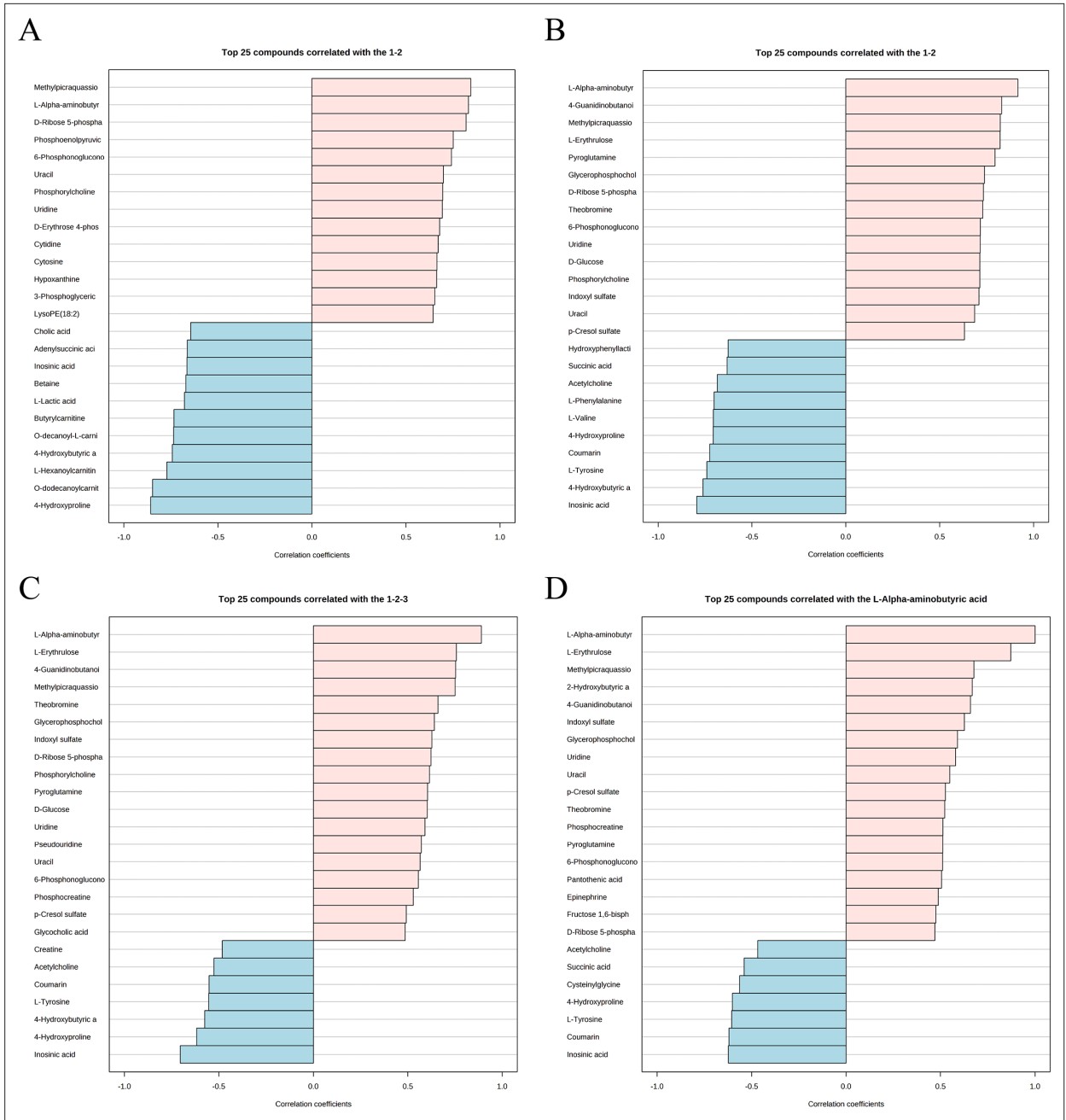

**Figure 10.** Expression patterns of metabolites. (**A**) Metabolite expression patterns associated with ovariectomy. (**B**) Metabolite expression patterns related to estrogen supplementation. (**C**) Metabolite expression patterns associated with increased estrogen concentration. (**D**) Metabolite expression patterns associated with L-AABA.

specifically conducted rat-specific metabolic pathway analysis, which showed significant differences in the Pentose phosphate pathway, Glycerophospholipid metabolism, Arginine and proline metabolism, and Pyrimidine metabolism (*Figure 12D*, *Supplementary file 29* : Rat-specific metabolic pathway analysis details).

## Discussion

The primary objective of this study is to explore the correlation and potential mechanisms between low estrogen status and postmenopausal hypertension through metabolomics analysis. The modeling in this study is consistent with previous research results, where estrogen levels significantly decrease

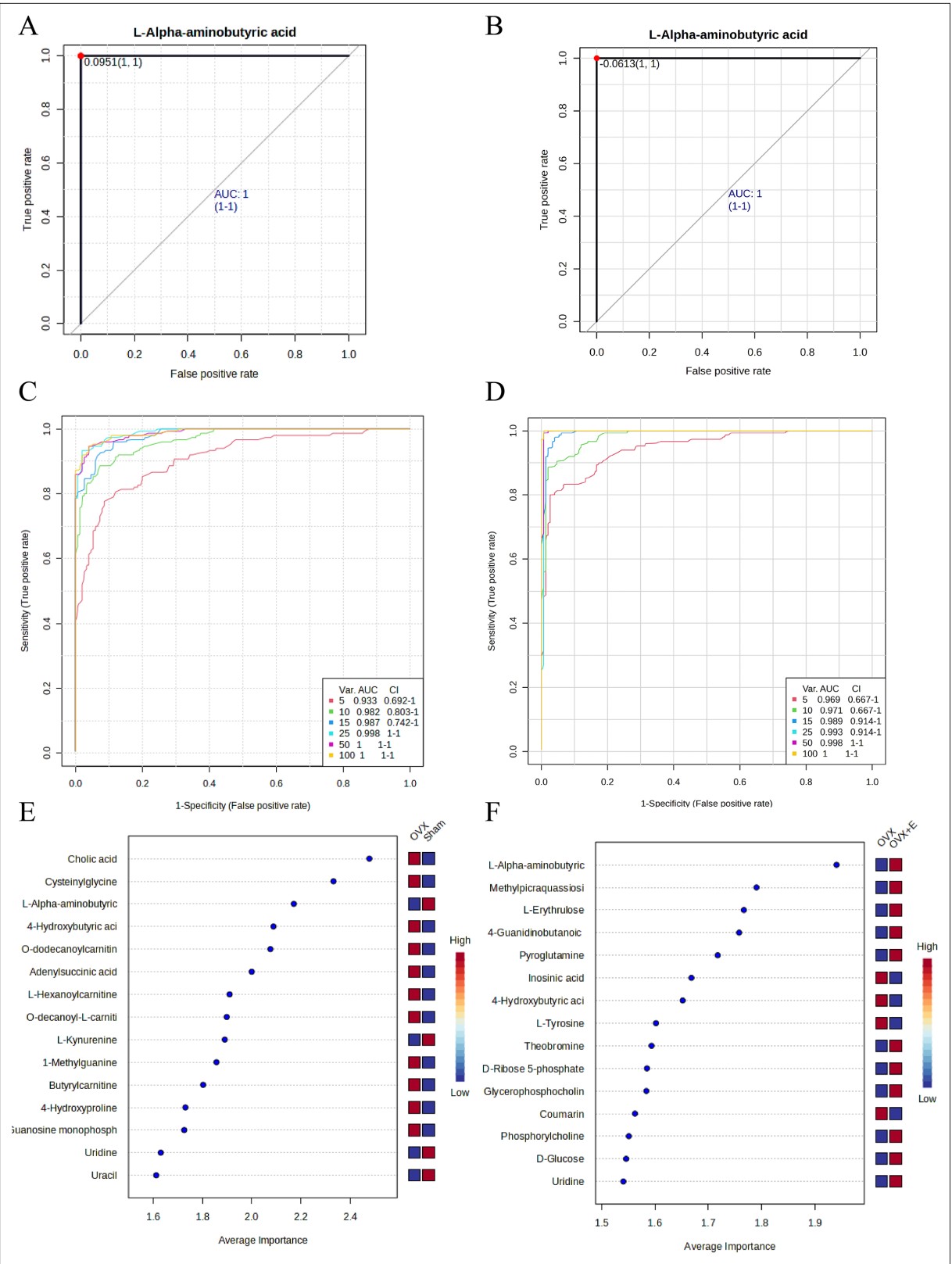

**Figure 11.** Screening of typical biomarkers. (**A**) ROC curve of L-AABA between the Sham group and OVX group. (**B**) ROC curve of L-AABA between the OVX group and OVX +E group. (**C**) Multivariate ROC curve based exploratory analysis between the Sham and OVX group. (**D**) Multivariate ROC curve based exploratory analysis between the OVX group and OVX +E group. (**E**) Average importance ranking of multivariate models for biomarkers between the Sham and OVX group. (**F**) Average importance ranking of multivariate models for biomarkers between the OVX and OVX +E group.

*Figure 11 continued on next page*

*Figure 11 continued*

The online version of this article includes the following figure supplement(s) for figure 11:

**Figure supplement 1.** The error classification of the multivariate ROC curve.

after ovariectomy, and exogenous estrogen supplementation can restore estrogen to physiological levels. Correspondingly, the SBP, DBP, and PP of the OVX group are significantly higher than those of the control group (Sham group), while after estrogen supplementation (OVX + E group), the above blood pressure parameters can return to the level of the control group (*Pitha et al., 2023*). The phenotypic changes mentioned above support further exploration of the mechanisms related to estrogen and hypertension.

Through various statistical analysis methods, we attempt to identify the most important mediator between estrogen and postmenopausal hypertension from a large number of differential metabolites. After extensive screening, we believe that L-AABA is the most promising biomarker. L-AABA plays a crucial role in certain metabolic pathways in the human body and can be used as a supplement to support overall health (*Wang et al., 2020*). It is an optically active form of α-aminobutyric acid (AABA), a non-essential amino acid primarily derived from the breakdown metabolism of methionine,

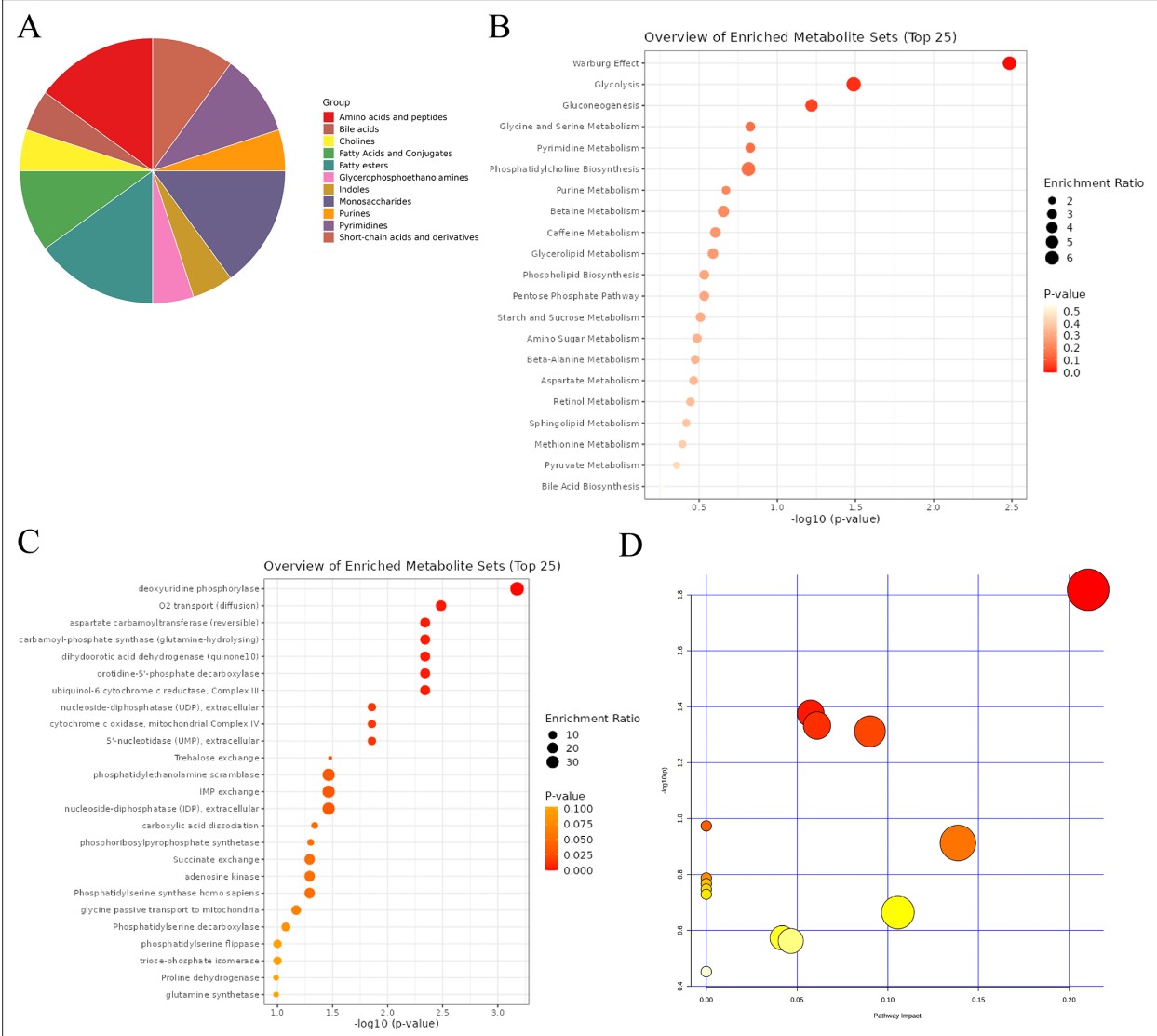

**Figure 12.** Comprehensive analysis of differential metabolites. (**A**) Classification pie chart of differential metabolites. (**B**) Metabolic enrichment analysis of differential metabolites. (**C**) Enzyme enrichment analysis of differential metabolites. (**D**) Enrichment analysis of rat-specific metabolite pathways.

threonine, and serine (*Haschke-Becher et al., 2016*). It serves as a biomarker for various diseases such as esophageal variceal bleeding (*Ai et al., 2022*) and septic liver injury (*Rudnick et al., 2009*). Recent studies have linked AABA to female osteoporosis (*Panahi et al., 2022*), a common condition in postmenopausal women, similar to postmenopausal hypertension. Our research found a significant downregulation of AABA in the aortic tissues of the OVX group, and estrogen supplementation could reverse the decrease in AABA levels, indicating that AABA may act as a protective factor in postmenopausal hypertension.

Based on our knowledge, AABA plays an important role in cardiovascular diseases: elevated levels of AABA were found in the heart tissues of dilated cardiomyopathy Syrian hamster models (*Maekawa et al., 2013*), significant increase of serum AABA in patients with atrial septal defect compared to healthy volunteers, and the ability of AABA levels to decrease to that of healthy volunteers after ductal occlusion, animal experiments have confirmed that AABA can prevent doxorubicin-induced cardiomyopathy in mice by increasing circulation and myocardial glutathione levels (*Irino et al., 2016*). It is worth noting that there has been no research exploring the correlation and potential mechanisms of AABA with hypertension. Current studies believe that high-intensity interval exercise has a corrective effect on hypertension (*Sosner et al., 2019*), and research has found that high-intensity interval exercise can increase plasma AABA concentrations (*Borges et al., 2024*), consistent with the trend of our experiments, suggesting that AABA may be a protective factor for hypertension. Although it is currently unclear whether AABA is involved in the improvement of hypertension, existing research results are enough to provide some hints for future research: vascular remodeling is a classic change in the pathophysiological process of hypertension and early vascular remodeling allows vessels to adapt to transient changes in hemodynamics and play compensatory protective effects. However, when blood pressure continues to increase, vessels cannot continue to compensate and undergo adverse restructuring, ultimately leading to a vicious cycle of lumen narrowing and rising blood pressure (*Gibbons and Dzau, 1994*). Although the related mechanisms of vascular remodeling in hypertensive patients have not been fully elucidated, current research suggests that the accumulation of macrophages in the vascular lumen and the inflammation resulting from it play an important role in the progression of vascular remodeling (*Suzuki et al., 2022*; *Zhou et al., 2010*; *Bakker et al., 2006*). It is worth noting that recent studies have confirmed that AABA can regulate the polarization and function of M1 macrophages by promoting oxidative phosphorylation and inhibiting glycolysis through the Warburg effect and gluconeogenesis (*Li et al., 2023*). Our rat-specific pathway enrichment results also coincidentally enriched in three important metabolic pathways: Warburg effect, glycolysis, and gluconeogenesis. Based on the above experimental results, we propose the following hypothesis: the low estrogen state caused by menopause leads to a decrease in L-AABA in the aorta, mediates macrophage activation, causes vascular remodeling, and ultimately leads to the occurrence of postmenopausal hypertension. Our team is conducting further research to verify the above points.

Enrichment analysis results indicate that the top three differential metabolites are amino acids and peptides, fatty acids and conjugates, and monosaccharides, which is in line with expectations as sugars, fats, and proteins are the three main types of metabolites in the human body. To avoid biases in research results caused by species differences, we further analyzed the specific metabolic pathways in rats, and the results show that the Warburg effect, glycolysis, and gluconeogenesis are the most enriched metabolic pathways. This suggests that sugar metabolism is the most significantly affected aspect of aortic metabolism in a low estrogen state. Current research generally believes that enhanced Warburg effect and glycolysis-mediated pulmonary vasoconstriction are key mechanisms in the development of pulmonary arterial hypertension (*Rafikov et al., 2015*; *Dunham-Snary et al., 2017*). Therefore, it is reasonable to speculate that changes in aortic Warburg effect and glycolysis may be involved in the occurrence of Postmenopausal Hypertension. Furthermore, researchers from the United States have found that the main culprit mediating the Warburg effect in tumors is macrophages in tumor tissue (*Reinfeld et al., 2021*). Whether aortic remodeling in postmenopausal women is also mediated by macrophage glucose metabolism reprogramming is a research direction worthy of exploration. Finally, in the exploration of key enzymes, we found deoxyuridine phosphorylase and $O_2$ transport (diffusion) as the top two candidates. In pancreatic cancer, UPP1 mediates the redox balance, survival, and proliferation of tumor cells under hypoxic conditions (*Nwosu et al., 2023*), and considering the enrichment of $O_2$ transport-related enzymes, the UPP-mediated hypoxic adaptation mechanism may also be an important participant in Postmenopausal Hypertension.

The current study is an exploratory work on the relationship between low estrogen and Postmeno-pausal Hypertension. Although some meaningful findings have been obtained, there are still some aspects that need further improvement. Firstly, there are several ways to construct animal models of menopause, including X-ray induced ovarian injury animal model (*Ling et al., 2017*), ovariectomized animal model (*da Rocha et al., 2012*), and chemical-induced ovarian failure model (*Van Kempen et al., 2011*). Among them, the ovariectomy animal model is the most commonly used at present, but it has limitations in simulating the drastic fluctuations in estrogen levels after ovariectomy. Secondly, the hypothesis that L-AABA regulates macrophage function to mediate blood pressure regulation has not been experimentally confirmed. Lastly, there is a lack of direct evidence of metabolic changes from human samples.

## Materials and methods

### Animal models

The experimental animals for this study were purchased from the Department of Laboratory Animal Science at Peking University Health Science Center. The animals' care and handling followed the guidelines established by the Animal Experimental Control and Supervision Committee, along with the Declaration of Helsinki by the World Medical Association regarding ethical standards for medical research involving animals. Approval for the experimental protocol was obtained from the Laboratory Animal Welfare Ethics Committee (No. LA2018092). Twenty-four 12-week-old female Sprague Dawley rats (Without any genetic modification) of SPF grade were randomly assigned to three groups: the Sham surgery group (Sham), ovariectomy group (OVX), and ovariectomy group treated with estrogen (OVX +E). These animals were housed under controlled conditions, including temperature (22–26°C), humidity (50–60%), and a 12 hr light/12 hr dark cycle, provided with a non-soy diet and ad libitum access to water. Surgery was performed one week after acclimatization. The Sham group underwent skin incision and closure, while the OVX and OVX +E groups underwent ovariectomy. On the 14th day post-surgery, all rats received subcutaneous injections of specific drugs between 9am and 10am daily for four weeks. Rats in the OVX +E group were administered 17β-estradiol (25 µg/kg/day; Sigma, St. Louis, MO, USA) dissolved in ethanol and diluted with sterile sesame oil (10 mg/0.1 mL, 0.25 mL/kg; GLBIO, Montclair, California, USA). The other groups were given an equal volume of sterile sesame oil. Starting from the third day post-operation, vaginal exfoliated cells were smeared daily for 7 days consecutively to confirm the successful establishment of the model. Vaginal cells were collected using a cotton swab dampened in 0.9% saline, stained with hematoxylin and eosin, dehydrated with alcohol, clarified with xylene, and ultimately preserved with resin.

### Blood pressure measurement

After 4 weeks of ovariectomy, blood pressure was measured in 24 rats every night from 22:00 to 24:00 for the next seven days. The first 6 days were used for adaptation training to reduce the impact of the surgery and surroundings on blood pressure. After blood pressure values stabilized, data collection began on the seventh day. The CODA-HT6 non-invasive blood pressure system (Kent Scientific Corporation, CT, USA) was used for the measurements. The cuff was positioned 1 cm from the tail base, attaching the VPR sensor to the tail. The tubing and VPR sensor were gradually inflated until the tail was completely blocked off, and then gradually deflated. Systolic blood pressure (SBP) was measured when blood flow was detected in the artery. The peak slope of the blood pressure change as determined by VPR revealed the diastolic blood pressure (DBP). Pulse pressure (PP) is equal to SBP minus DBP. Every rat was subjected to ten to fifteen measurement cycles; the average of the last five cycles was taken as the final result.

### Harvesting aorta and blood samples

All rats were killed after having their blood pressure measured and given an intraperitoneal injection of 1% pentobarbital sodium (80 mg/kg; Sigma, St. Louis, MO, USA). After drawing blood samples from the heart, they were centrifuged at 4 °C. A cold 0.9% saline solution was utilized to perfuse the heart prior to the collection of aortic tissues. After that, the whole aorta and the serum fraction were kept in storage at –80 °C.

### Radioimmunoassay

With a detection limit of 3 pg/mL, the Rat E2 ELISA kit (RE1649-48T, Bioroyee, Beijing, China) was used for radioimmunoassay to evaluate serum estrogen levels. Samples were separated, centrifuged, and then incubated for analysis in accordance with the instructions.

### Preparing the aorta for metabolomics

Precisely weigh out 20±1 mg of aorta tissue, and then combine it with 500 µL of methanol containing an internal standard of 5 µg/mL 2-chloro-l-phenylalanine. The mixture was homogenized for 90 s using a high-throughput tissue grinder (60 Hz; Tissuelyser-24, Jingxin, Shanghai, China). A 15-min centrifugation at 12,000 rpm and 4 °C resulted in the separation of 100 µL of the supernatant for analysis using metabolomics.

### Metabolomics measurement

The UHPLC-Q-TOF technique was used to analyze the metabolomics of aorta tissue. ACQUITY UPLC HSS T3 columns (1.8 µm, 2.1 mm × 100 mm, Waters, Dublin, Ireland) were used as the chromatographic column in the Agilent 1290 II UPLC-QTOF 5600 PLUS (Sciex) liquid chromatography-mass spectrometry system, which was used in the electric spray ionization (ESI) mode. Curtain gas = 35, ion spray voltage = 5500 V (positive ion mode) and –4500 V (negative ion mode), temperature = 450 °C, ion source gas 1=50, and ion source gas 2=50 were the parameters for liquid chromatography-mass spectrometry. Software for Agilent MassHunter workstations (version B.01.04; Agilent, Lexington, MA, USA) was used to process raw data. By increasing the intensity threshold to 300, noise was filtered out and isotope interference was eliminated. The method of identifying metabolite was by comparison with the publicly available METLIN database (here, access date: 9 October 2021).

### Data preprocessing and bioinformatics analysis

In the current research, data preprocessing and bioinformatics analysis were conducted using MetaboAnalyst 5.0 (*Pang et al., 2021*) (http://www.metaboanalyst.ca/; visited on October 12, 2021). The Sham group served as the reference group while the data were normalized using the group probability quotient normalization technique (*Dieterle et al., 2006*). Data normalization was accomplished using log transformation (base 10) and the Pareto approach (mean-centered, divided by the square root of each variable's standard deviation). Several analytical techniques were used for information mining, such as the Debiased Sparse Partial Correlation (DSPC) network (*Basu et al., 2017*; *Figure 1*) summarizes the aforementioned study procedure.

### Conclusion

This study for the first time delineated the metabolic characteristics of the aorta under low estrogen status and explored potential mechanisms of postmenopausal hypertension. Sugar metabolism reprogramming plays an important role in Postmenopausal Hypertension, and AABA may be a key link in the pathogenic mechanism. The aforementioned mechanisms may be the future focus of work on Postmenopausal Hypertension and deserve further in-depth exploration.

## Acknowledgements

The authors would like to thank Professor Qin Lihua's research team at the Peking University Health Science Center for providing animal research support. The authors declare financial support was received for the research, authorship, and/or publication of this article. This work was funded by Natural Science Foundation of Shandong Province (ZR2024QH262, ZR2024QH640).The Youth Research Fund of Qingdao University Affiliated Hospital (QDFYQN2023111, QDFYQN2023119), Natural Science Foundation of Qingdao Municipality (24-4-4-zrjj-105-jch).

# Additional information

## Funding

| Funder | Grant reference number | Author |
| --- | --- | --- |
| Natural Science Foundation of Shandong Province | ZR2024QH262 | Yao Li |
| The youth research found of the affiliated hospital of Qingdao university | QDFYQN2023119 | Yao Li |
| Natural Science Foundation of Shandong Province | ZR2024QH640 | Wei Zhang |
| The youth research found of the affiliated hospital of Qingdao university | QDFYQN2023111 | Wei Zhang |
| Natural Science Foundation of Qingdao Municipality | 24-4-4-zrjj-105-jch | Wei Zhang |

The funders had no role in study design, data collection and interpretation, or the decision to submit the work for publication.

## Author contributions

Yao Li, Conceptualization, Data curation, Formal analysis, Funding acquisition, Visualization, Methodology, Writing - original draft, Writing – review and editing; Hui Xin, Methodology, Writing - original draft, Writing – review and editing; Zhexun Lian, Supervision, Project administration, Writing – review and editing; Wei Zhang, Conceptualization, Supervision, Validation, Project administration, Writing – review and editing

## Author ORCIDs

Wei Zhang https://orcid.org/0000-0002-3167-0002

## Ethics

The experimental animals for this study were purchased from the Department of Laboratory Animal Science at Peking University Health Science Center. The animals' care and handling followed the guidelines established by the Animal Experimental Control and Supervision Committee, along with the Declaration of Helsinki by the World Medical Association regarding ethical standards for medical research involving animals. Approval for the experimental protocol was obtained from the Laboratory Animal Welfare Ethics Committee (No. LA2018092).

Reviewer #1 (Public review): https://doi.org/10.7554/eLife.101701.2.sa1
Reviewer #2 (Public review): https://doi.org/10.7554/eLife.101701.2.sa2
Reviewer #3 (Public review): https://doi.org/10.7554/eLife.101701.2.sa3
Author response https://doi.org/10.7554/eLife.101701.2.sa4

# Additional files

## Supplementary files

Supplementary file 1. Metabolomics raw data.

Supplementary file 2. Non-parametric tests of metabolites.

Supplementary file 3. Fisher's LSD tests of metabolites.

Supplementary file 4. Intersection differential metabolites.

Supplementary file 5. Differential metabolites identified by SAM.

Supplementary file 6. The correlation coefficient and p-value of the correlation analysis for all

metabolites.

Supplementary file 7. The correlation coefficient and p-value of the correlation analysis for all metabolites.

Supplementary file 8. The correlation coefficient and p-value of the correlation analysis for all samples.

Supplementary file 9. The correlation coefficient and p-value of the correlation analysis for all samples.

Supplementary file 10. PLS-DA cross validation details.

Supplementary file 11. P scores for differential metabolites.

Supplementary file 12. Mean Metabolic Accuracy of differential metabolites in random forest tree model.

Supplementary file 13. Details of t-test for differential metabolites between Sham group and OVX group.

Supplementary file 14. Details of t-test for differential metabolites between OVX +E group and OVX group.

Supplementary file 15. VIP scores of OPLS-DA models between subgroups.

Supplementary file 16. VIP scores of OPLS-DA models between subgroups.

Supplementary file 17. VIP scores of the random forest model between subgroups.

Supplementary file 18. VIP scores of the random forest model between subgroups.

Supplementary file 19. The evaluation details of EBAM and SAM model between subgroups.

Supplementary file 20. e evaluation details of EBAM and SAM model between subgroups.

Supplementary file 21. The evaluation details of EBAM and SAM model between subgroups.

Supplementary file 22. The evaluation details of EBAM and SAM model between subgroups.

Supplementary Pistea 23. Details of intergroup differential analysis of low estrogen related metabolites.

Supplementary file 24. Details of intergroup differential analysis of low estrogen related metabolites.

Supplementary file 25. Details of metabolite related trends from low to high estrogen concentrations.

Supplementary file 26. Correlation coefficient of metabolites related to L-AABA expression.

Supplementary file 27. The promising differential metabolites AUC results of ROC curves between subgroups.

Supplementary file 28. The promising differential metabolites AUC results of ROC curves between subgroups.

Supplementary file 29. Rat-specific metabolic pathway analysis details.

Supplementary file 30. The legends for the figures.

MDAR checklist

Source data 1. Metabolomics data of rat aorta.

## Data availability

The original contributions presented in the study are included in *Supplementary files 1–29*.

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
