## [Editor Report · eLife Assessment]

This **useful** study provides **incomplete** evidence regarding the pathophysiological role of low estrogen levels post-menopause in hypertension, focusing on L-AABA as a key mediator. The results describe a novel hypothesis for the pathophysiology of hypertension in this population and are of interest to experts in hypertension and vascular biology.

---

## [Referee Report · Reviewer #1 (Public review)]

The authors aim to investigate the relationship between low estrogen levels, postmenopausal hypertension, and the potential role of the molecule L-AABA as a biomarker for hypertension. By employing metabolomic analysis and various statistical methods, the study seeks to understand how estrogen deficiency affects blood pressure and identify key metabolites involved in this process, with a particular focus on L-AABA.

Strengths:

The study addresses a relevant and understudied area: the role of estrogen and metabolites in postmenopausal hypertension. It presents a novel hypothesis that L-AABA may serve as a protective factor against hypertension, which could have significant clinical implications if proven.

Weaknesses:

The evidence linking L-AABA to hypertension is largely correlative, lacking experimental validation or mechanistic proof. Key limitations, such as the inadequacy of the ovariectomy model in replicating human menopause, are acknowledged but not addressed with alternative approaches. In summary, while the study offers an intriguing hypothesis, its conclusions are premature and require further experimental validation and human data to substantiate the claims.

---

## [Referee Report · Reviewer #2 (Public review)]

Summary:

In this manuscript, Dr. Yao Li et al. documented the metabolomic profile of the aorta from OVX rats and that from OVX plus E2. These conditions mimic post-menopause hypertension and hormonal replacement therapy.

Strengths:

The authors state that this is probably the first study to examine the metabolic changes in the aorta of post-menopause hypertension.

Weaknesses:

There are several weaknesses, and a few of them are quite serious.

(1) The aorta is not a resistant artery and has little to do with hypertension. The authors should have used resistant arteries for this study. The expression of several adrenergic receptors and cholinergic receptors in the aorta and resistant arteries are different. It is unknown whether the aorta metabolomic profile has any relevance to BP and whether they are similar to that of the resistant arteries. I understand the logistics issue of obtaining enough tissues from resistant arteries. At least, once some leads are discovered in the aorta, the authors should validate it in resistant arteries. This should be feasible.

(2) The aorta and all the arteries have three layers. It is critically important to know whether the metabolic changes occur in the intima or in the media, while the adventitia probably has little to do with vasoconstriction and hypertension. If the authors want to use the aorta to conduct the preliminary study, they should completely remove the adventitia and then use samples with and without their endothelium stripped and then assess their metabolomic profiles. After the leads are obtained from this preliminary profiling, they should be validated in endothelium and smooth muscles of the resistant artery. The current experiments are not appropriately designed.

(3) The tail-cuff BP measurement is a technique of the last century. The current gold standard of BP measurement is by telemetry. The tail-cuff method is particularly problematic in this study because the 1-2 h restraining of the rats for more than 10 times BP measurement will cause significant stress in the animal, and their stress hormone secretion might cause biased metabolomic profiles in the OVX versus shames operated mice. The problem can be totally avoided by using telemetry.

(4) Although the L-AABA showed a high p-value (10^-4) of a decrease in the OVX rats, the fold change is small (2-3 folds). Such a small change should be validated using a different method to be convincing.

(5) The authors claim (or hypothesize) that the reduced AABA level in OVX can cause vascular remodeling. This can be easily validated by the histology of the OVX-resistant artery, and they should do that during the revision. The authors should also examine the M1 macrophage function from the OVX mice to validate their claimed link of AABA to M1.

(6) As mentioned above, the authors need to pinpoint the changes of AABA to target cells, i.e., endothelial cells, SMC, or M1, and then use in vitro or in vivo cell biology approaches to assess whether these cells in the OVX rat indeed have an abnormality in function and, indeed, such functional changes are responsible for the BP phenotype.

(7) The results of the current study can be condensed into 1 or 2 figures that can serve as a base or a starting point for a deeper scientific study.

Summary

The experimental design of this manuscript is inappropriate, and the methods are not up to the current standards. The whole study is descriptive and rudimentary. It lacks validation and mechanism. The data from this manuscript might be of some value and can serve as the first step for more investigation of the mechanism of post-menopause hypertension.

---

## [Referee Report · Reviewer #3 (Public review)]

Summary:

The decrease in estrogen levels is strongly associated with postmenopausal hypertension. Dr. Yao Li and colleagues aimed to investigate the metabolomic mechanisms of underlying postmenopausal hypertension using OVX and OVX+E2 rat models. They successfully established a correlation between reduced estrogen levels and the development of hypertension in rats. They identified L-alpha-aminobutyric acid (AABA) as a potential marker for postmenopausal hypertension. The research explored the metabolic alterations in aortic tissues and proposed several potential mechanisms contributing to postmenopausal hypertension.

Strengths:

The group performed a comprehensive enrichment analysis and various statistical analyses of the metabolomics data.

Weaknesses:

(1) The manuscript is descriptive in nature, although they mentioned their primary objective is to explore the potential mechanisms linking low estrogen levels with postmenopausal hypertension. No mechanism insights have been interrogated in this study, which has been mentioned by the authors in the discussion. The connection between E2, AABA, and macrophage needs to be validated in endothelial cells, vascular smooth muscle cells, and other aortic tissue cells. Without such verification, the manuscript predominantly raises hypotheses only based on metabolomic data.

(2) The serum contains three forms of estrogen: Estradiol, Estrone, and Estriol. The authors used the Rat E2 ELISA kit. Ideally, all three forms of estrogen should be measured.

---

## [Author Response]

**Reviewer #1 (Public review):**
Summary:The authors aim to investigate the relationship between low estrogen levels, postmenopausal hypertension, and the potential role of the molecule L-AABA as a biomarker for hypertension. By employing metabolomic analysis and various statistical methods, the study seeks to understand how estrogen deficiency affects blood pressure and identify key metabolites involved in this process, with a particular focus on L-AABA.Strengths:The study addresses a relevant and understudied area: the role of estrogen and metabolites in postmenopausal hypertension. It presents a novel hypothesis that L-AABA may serve as a protective factor against hypertension, which could have significant clinical implications if proven.

We appreciate the acknowledgment of our study’s focus on an important and understudied area. Our hypothesis regarding L-AABA’s role as a possible protective factor against hypertension indeed holds promise for advancing clinical implications.

Weaknesses:The evidence linking L-AABA to hypertension is largely correlative, lacking experimental validation or mechanistic proof. Key limitations, such as the inadequacy of the ovariectomy model in replicating human menopause, are acknowledged but not addressed with alternative approaches. In summary, while the study offers an intriguing hypothesis, its conclusions are premature and require further experimental validation and human data to substantiate the claims.

We recognize the limitations regarding the correlative nature of our findings and the inadequacy of the OVX model in replicating human menopause. Future research will prioritize experimental validation and incorporate human studies to solidify our conclusions.

**Reviewer #2 (Public review):**
Summary:In this manuscript, Dr. Yao Li et al. documented the metabolomic profile of the aorta from OVX rats and that from OVX plus E2. These conditions mimic post-menopause hypertension and hormonal replacement therapy.Strengths:The authors state that this is probably the first study to examine the metabolic changes in the aorta of post-menopause hypertension.

As pointed out by the reviewer, our study may be the first to investigate changes in aortic metabolism in postmenopausal hypertension. As an exploratory study, our goal is to depict the overall characteristics and explore possible research directions.

Weaknesses:There are several weaknesses, and a few of them are quite serious.(1) The aorta is not a resistant artery and has little to do with hypertension. The authors should have used resistant arteries for this study. The expression of several adrenergic receptors and cholinergic receptors in the aorta and resistant arteries are different. It is unknown whether the aorta metabolomic profile has any relevance to BP and whether they are similar to that of the resistant arteries. I understand the logistics issue of obtaining enough tissues from resistant arteries. At least, once some leads are discovered in the aorta, the authors should validate it in resistant arteries. This should be feasible.

We acknowledge the limitation of using the aorta and will aim to include studies on resistant arteries to validate our metabolomic findings.

(2) The aorta and all the arteries have three layers. It is critically important to know whether the metabolic changes occur in the intima or in the media, while the adventitia probably has little to do with vasoconstriction and hypertension. If the authors want to use the aorta to conduct the preliminary study, they should completely remove the adventitia and then use samples with and without their endothelium stripped and then assess their metabolomic profiles. After the leads are obtained from this preliminary profiling, they should be validated in endothelium and smooth muscles of the resistant artery. The current experiments are not appropriately designed.

Future studies will involve detailed profiling of specific arterial layers, focusing on the intima and media to enhance the relevance of our findings related to hypertension.

(3) The tail-cuff BP measurement is a technique of the last century. The current gold standard of BP measurement is by telemetry. The tail-cuff method is particularly problematic in this study because the 1-2 h restraining of the rats for more than 10 times BP measurement will cause significant stress in the animal, and their stress hormone secretion might cause biased metabolomic profiles in the OVX versus shames operated mice. The problem can be totally avoided by using telemetry.

We appreciate the suggestion and will consider telemetry for more accurate blood pressure measurements in future experiments to minimize stress-related bias.

(4) Although the L-AABA showed a high p-value (10^-4) of a decrease in the OVX rats, the fold change is small (2-3 folds). Such a small change should be validated using a different method to be convincing.

We plan to employ additional methods to validate the observed changes in L-AABA levels in the following research, ensuring robustness of our findings.

(5) The authors claim (or hypothesize) that the reduced AABA level in OVX can cause vascular remodeling. This can be easily validated by the histology of the OVX-resistant artery, and they should do that during the revision. The authors should also examine the M1 macrophage function from the OVX mice to validate their claimed link of AABA to M1.

We intend to conduct histological analyses and examine M1 macrophage function in OVX-resistant arteries to validate our hypothesis in the following research.

(6) As mentioned above, the authors need to pinpoint the changes of AABA to target cells, i.e., endothelial cells, SMC, or M1, and then use in vitro or in vivo cell biology approaches to assess whether these cells in the OVX rat indeed have an abnormality in function and, indeed, such functional changes are responsible for the BP phenotype.

Addressing these points, we aim to pinpoint specific cell types affected by AABA variations and conduct in vitro and in vivo studies to examine their physiological impacts in the following research.

(7) The results of the current study can be condensed into 1 or 2 figures that can serve as a base or a starting point for a deeper scientific study.

Thank you for your suggestion. As a omics research, our research approach may differ from traditional mechanism studies.

SummaryThe experimental design of this manuscript is inappropriate, and the methods are not up to the current standards. The whole study is descriptive and rudimentary. It lacks validation and mechanism. The data from this manuscript might be of some value and can serve as the first step for more investigation of the mechanism of post-menopause hypertension.
**Reviewer #3 (Public review):**
Summary:The decrease in estrogen levels is strongly associated with postmenopausal hypertension. Dr. Yao Li and colleagues aimed to investigate the metabolomic mechanisms of underlying postmenopausal hypertension using OVX and OVX+E2 rat models. They successfully established a correlation between reduced estrogen levels and the development of hypertension in rats. They identified L-alpha-aminobutyric acid (AABA) as a potential marker for postmenopausal hypertension. The research explored the metabolic alterations in aortic tissues and proposed several potential mechanisms contributing to postmenopausal hypertension.Strengths:The group performed a comprehensive enrichment analysis and various statistical analyses of the metabolomics data.

As summarized by the reviewer, our current study conducted a comprehensive analysis of metabolomics data. It is also a reliable foundation for further mechanism research.

Weaknesses:(1) The manuscript is descriptive in nature, although they mentioned their primary objective is to explore the potential mechanisms linking low estrogen levels with postmenopausal hypertension. No mechanism insights have been interrogated in this study, which has been mentioned by the authors in the discussion. The connection between E2, AABA, and macrophage needs to be validated in endothelial cells, vascular smooth muscle cells, and other aortic tissue cells. Without such verification, the manuscript predominantly raises hypotheses only based on metabolomic data.

We have proposed research hypotheses based on detailed omics data. Further research on the mechanisms involving endothelial and vascular smooth muscle cells to validate the pathway connections between E2, AABA, and macrophages is undoubtedly the future direction of this study.

(2) The serum contains three forms of estrogen: Estradiol, Estrone, and Estriol. The authors used the Rat E2 ELISA kit. Ideally, all three forms of estrogen should be measured.

Future assays will aim to measure Estradiol, Estrone, and Estriol to capture a more comprehensive picture of estrogen’s role in postmenopausal hypertension.